# Understanding Gradient Descent through the Training Jacobian

## Abstract

We examine the geometry of neural network training using the Jacobian of trained network parameters with respect to their initial values. Our analysis reveals low-dimensional structure in the training process which is dependent on the input data but largely independent of the labels. We find that the singular value spectrum of the Jacobian matrix consists of three distinctive regions: a "chaotic" region of values orders of magnitude greater than one, a large "bulk" region of values extremely close to one, and a "stable" region of values less than one. Along each bulk direction, the left and right singular vectors are nearly identical, indicating that perturbations to the initialization are carried through training almost unchanged. These perturbations have virtually no effect on the network's output in-distribution, yet do have an effect far out-of-distribution. While the Jacobian applies only locally around a single initialization, we find substantial overlap in bulk subspaces for different random seeds.

## 1 Introduction

Prior work suggests that neural network training is in some sense "intrinsically low-dimensional." For example, Li et al. (2018) show that projecting gradient descent updates onto a randomly oriented low-dimensional subspace has minimal impact on the final network's performance. For problems like image classification on MNIST or CIFAR-10, the intrinsic dimensionality appears to be in the hundreds or low thousands.

Relatedly, Gur-Ari et al. (2018) find that during SGD training, the stochastic gradient is approximately contained in the dominant eigenspace of the Hessian matrix, a space of dimensionality roughly equal to the number of classes (which may be as low as ten). However, Song et al. (2024) find that restricting training to the dominant Hessian eigenspace causes the loss to plateau at a high level, while training in the complement of this subspace performs as well as standard training. Additionally, the effect noted by Gur-Ari et al. seems to disappear as the batch size increases, indicating that it is an artifact of minibatch noise. This cautionary tale shows that different ways of operationalizing the "dimensionality" of training can yield different results.

In this work, we propose a new way of approaching this question, based on linearizing training around a given initialization. For small networks, we are able to differentiate through the entire training process, obtaining the Jacobian of final parameters with respect to initial ones. The singular value decomposition of this matrix specifies a basis for initial parameters, and another basis for final parameters, such that training can be locally approximated as an affine map that stretches or shrinks parameter space along these two bases.

We find that the singular value spectrum of the Jacobian consists of three distinctive regions: a "chaotic" region of values orders of magnitude greater than one, a large "bulk" region of values extremely close to one, and a "stable" region of values less than one. We find that this bulk depends only weakly on the initialization point, loss function, and labels, but depends strongly on the data distribution; in particular, random data does not have a bulk.

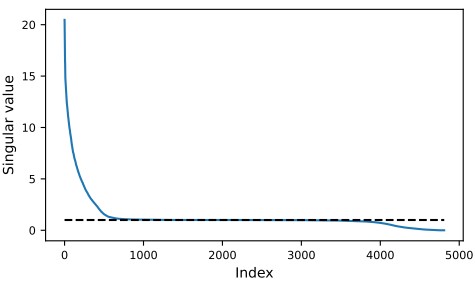
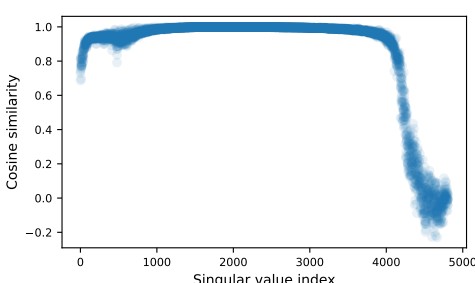

(a) Singular value spectrum, with a black dashed line highlighting the singular values near one.

(b) Cosine similarity of corresponding left and right singular vectors for the training Jacobian.

Figure 1: Spectral analysis of the training Jacobian for an MLP with a single hidden layer, trained using SGD with momentum for 25 epochs until training loss reached zero.

## 2 LINEARIZING TRAINING

Let $\theta_0 \in \mathbb{R}^N$ be an initial parameter vector and let $\mathcal{B} \subset \mathbb{R}^N$ be a small ball of radius $\epsilon$ centered at $\theta_0$. Now consider how $\mathcal{B}$ as a whole is warped by the training function $f : \mathbb{R}^N \to \mathbb{R}^N$. For sufficiently small $\epsilon$, the training process is well-approximated by its first order Taylor series around $\theta_0$,

$$f(\theta) \approx f(\theta_0) + J(\theta_0)(\theta - \theta_0) \tag{1}$$

where $J(\theta_0)$ is the Jacobian of the trained parameters with respect to their initial values, a matrix we will call the **training Jacobian**. The image of $\mathcal{B}$ is then an ellipsoid whose principal axes are the left singular vectors of $J(\theta_0)$, and whose radii are proportional to the corresponding singular values.

For a simple example, let's consider the case of a quadratic objective $\mathcal{L}(\theta) = \frac{1}{2}\theta^T \mathbf{H}\theta$ in the gradient flow limit. Then the parameters at any time $t$ are a linear function of the initial parameters: $f(\theta_0, t) = J_t(\theta_0) = \exp(-\mathbf{H}t)\theta_0$. Trivially, if the Hessian $\mathbf{H}$ is positive definite, then the training Jacobian tends to the zero matrix as $t \to \infty$, since there is a unique minimum at $\theta = 0$.[1] In this case, the "dimensionality of training" is clearly equal to $N$, the full dimensionality of parameter space.

But in the loss landscapes of deep neural networks, the Hessian is known to be highly singular (Sagun et al., 2017; Ghorbani et al., 2019; Yao et al., 2020). In the toy quadratic model above, each zero eigenvalue of $\mathbf{H}$ corresponds to a singular value of $\exp(0) = 1$ in the training Jacobian. Along these zero curvature directions, the parameters do not change at all, and $\mathcal{B}$ retains its original radius $\epsilon$. Clearly, training "does not happen" along this subspace, and the dimensionality of training is equal to $\mathrm{rank}(\mathbf{H}) = \mathrm{rank}(J_t(\theta_0) - \mathbf{I})$.

For more sophisticated objectives and optimizers, the quantity $\mathrm{rank}(J_t(\theta_0) - \mathbf{I})$ continues to be well-defined, while $\mathrm{rank}(\mathbf{H})$ does not since the Hessian varies across the training trajectory. It measures the dimension of the subspace along which a small ball centered around the initialization retains its original radius— that is, the subspace along which *volumes* are locally preserved.

## 3 EXPERIMENTS

We use the forward-mode automatic differentiation functionality in JAX (Bradbury et al., 2018) to efficiently compute the training Jacobian for small networks.[2] For computational tractability, we train MLPs with one hidden layer of width 64 on the UCI digits dataset (Alpaydin & Kaynak, 1998) included in Scikit Learn (Pedregosa et al., 2011) in most of our experiments. We train for 25 epochs, which is sufficient to achieve near-zero training loss. See Appendix A.1 for a partial extension of these experiments to network over ten times larger.

---

[1]This is more generally true for any strongly convex objective with an appropriately tuned optimizer.

[2]Specifically, we use `jax.jacfwd`, which has memory complexity constant in the number of training steps, yet quadratic in the number of parameters. Alternatively we can compute the Jacobian row-by-row using `jax.linearize`, whose memory usage is linear in both the parameter count and number of training steps.

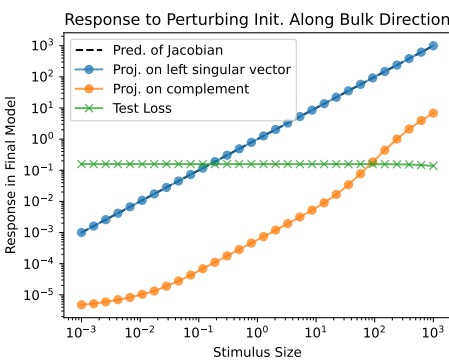 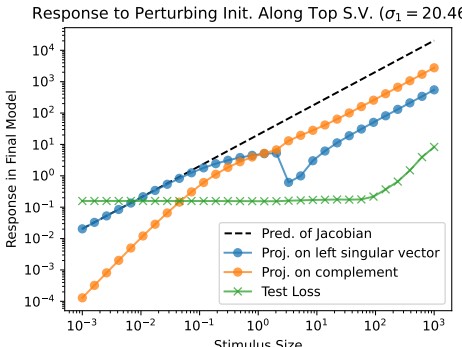

(a) Effect of perturbing the initialization along a randomly selected bulk singular vector. The response in the trained model matches the Jacobian "prediction" across seven orders of magnitude.

(b) Effect of perturbing the initialization along the top singular vector. Unlike the bulk direction, training leaves the linear regime between $\lambda = 10^{-1}$ and $\lambda = 10^0$.

Figure 2: The linearization of training remains valid for much longer along bulk directions than it does along a top singular vector. The orange line indicates the Euclidean norm of the response projected onto the orthogonal complement of the span of the singular vector; if training is purely linear, this quantity should be zero.

### 3.1 Spectral Analysis of the Jacobian

The singular value spectrum of the training Jacobian is shown in Figure 1a. Strikingly, about 3000 out of 4810 singular values are extremely close to one. Furthermore, the left and right singular vectors corresponding to these singular values are nearly identical (Figure 1b). This indicates that local perturbations along these directions, making up nearly two-thirds of the dimensionality of parameter space, are carried through training virtually unchanged. In what follows, we will refer to the span of these singular vectors as the **bulk subspace**, or simply the "bulk."

The first 500 or so singular values are also notable. Along these directions, perturbations to the initial parameters are *magnified* by training. These are due to the nonconvexity of the objective: gradient descent on a convex objective with an appropriate step size will always have a spectral norm no greater than one. Interestingly, while left and right singular vectors are not identical in this part of the spectrum, they still have cosine similarity in excess of 0.6, which is very high for vectors of this dimensionality. We will refer to the span of these singular vectors the **chaotic subspace**.

Finally, the last 750 or so singular vectors are less than one. Along this **stable subspace**, perturbations to the initial parameters are partially canceled out by the optimizer.

### 3.2 How Linear is Training?

In this section, we empirically evaluate how far we can perturb $\theta_0$ before the linearization of training in Eq. 1 breaks down. Specifically, for each singular value $\sigma_i$, we perform a line search with stimulus sizes $\lambda$ selected from a log-spaced grid ranging from $10^{-3}$ to $10^3$. For each value of $\lambda$, we run training on $\theta_0 + \lambda v_i$, where $v_i$ is the $i^{\text{th}}$ right singular vector. We record the delta between original and perturbed final parameters $\delta = f(\theta_0 + \lambda v_i) - f(\theta_0)$, and project this difference onto the $i^{\text{th}}$ left singular vector $u_i$. If we are in a regime where training is roughly linear, this quantity (the "response") should be close to $\lambda \sigma_i$.

We plot results for a representative bulk direction and the top singular vector in Figure 2. Remarkably, the responses (in blue) almost perfectly line up with what the linearized approximation of training would predict (dashed line) across *seven orders of magnitude*, up to $10^3$.

Our linear model of training also predicts that perturbations along the right singular vector $v_i$ should be "surgical," in the sense that they have zero effect on the final parameters along any direction orthogonal to the left singular vector $u_i$. We test this prediction by plotting the Euclidean norm

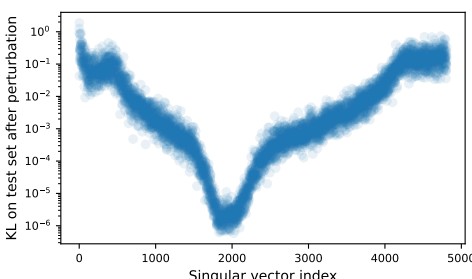

(a) Perturbations along bulk directions have virtually no effect on test set predictions, while perturbations along the chaotic and stable subspaces do have a notable effect.

(b) When evaluating the model on white noise images, perturbations along bulk directions have roughly the same effect size as perturbations along other subspaces.

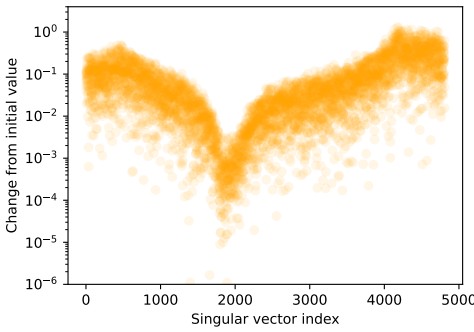

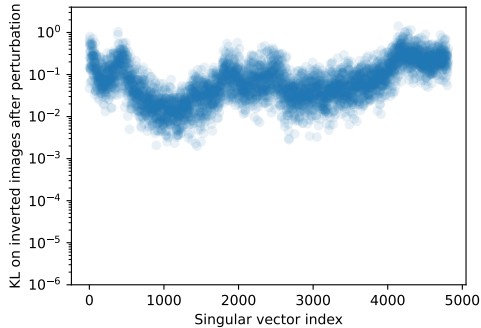

(c) Parameters change little along the bulk subspace compared to the chaotic and stable subspaces.

(d) When evaluating on images with inverted colors, perturbations along the bulk have at least as much impact as perturbations along other subspaces.

Figure 3: While the bulk has virtually no effect on in-distribution behavior, it does affect predictions far out-of-distribution (Panels b and d).

of the response projected onto $\mathrm{span}(u_i)^\perp$ on the orange line. If training were perfectly linear, this quantity would be zero. In practice, we find it stays remarkably small even for large stimulus sizes.

The story is quite different for perturbations along the top singular vector (Figure 2b). The perturbation rapidly becomes non-surgical as the stimulus size increases from $10^{-3}$ to $10^{-1}$, with the projection onto $\mathrm{span}(u_i)^\perp$ (orange) overtaking the projection onto $\mathrm{span}(u_i)$ (blue) when $\lambda > 1$.

### 3.3 DOES THE BULK MATTER?

In this experiment, we measure how much a perturbation of the final network weights along each of the Jacobian singular vectors affects the network's predictions on a held-out i.i.d. test set. Specifically, we compute the average KL divergence

$$\mathbb{E}_x\big[D_{\mathrm{KL}}(g(x;\theta)||g(x;\theta + u_i))\big], \tag{2}$$

where $g(x;\theta)$ denotes the network's output on input $x$ given parameters $\theta$, and $u_i$ is the $i^{\mathrm{th}}$ right singular vector of the training Jacobian.

Results are shown in Figure 3. For comparison, we also run the experiment on two far out-of-distribution datasets: a copy of the test set with all pixels inverted, and a set of uniform white noise images. We find that bulk directions have a much larger behavioral effect on far OOD behavior than they do on the test set.

These results make it fairly intuitive why the bulk subspace is left virtually unchanged by SGD: it simply does not affect the network's output on the training data distribution.

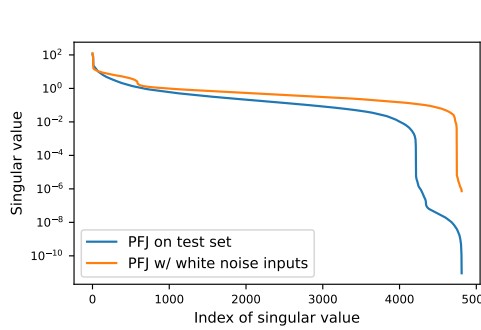

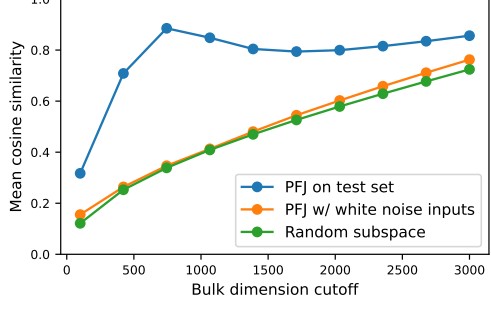

(b) Mean cosine of principal angles between the training-Jacobian bulk and the approximate nullspace of the PFJ. For each cutoff $k$, we compare the span of the $k$ right singular vectors of the PFJ with smallest singular value to the span of the $k$ right singular vectors of the training Jacobian with singular value closest to 1.

(a) Singular values of the PFJ for the digits test set and for a dataset of random noise. The sharp dropoff in singular value for the test set occurs before the last 597 values; for the white-noise dataset there is a dropoff before the last 65 values.

Figure 4: The parameter-function Jacobian on test images has a fairly large approximate nullspace, which is close to the training-Jacobian bulk. This effect disappears for white-noise images.

### 3.4 Relationship with the Parameter-function Jacobian

To complement the above experiments, we investigate the effect of bulk directions on network behavior by comparing the training Jacobian to the the Jacobian of network predictions (log-probabilities) on an entire dataset with respect to network parameters. We will refer to this as the parameter-function Jacobian (PFJ).

We compute the PFJ on a randomly initialized model, for two different sets of inputs: the digits test set and a set of white noise images. As shown in (Figure 4a), the PFJ for the test set has a distinct cluster of approximately 600 extremely-small singular values, indicating a large number of parameter-space directions that do not significantly affect the model's predictions. For the white noise dataset, this cluster is absent.

For the smallest singular values, we compare the corresponding singular vectors (the approximate nullspace) to the bulk subspace from the training Jacobian (Figure 4b). The nullspace for the digits dataset overlaps strongly with the training Jacobian bulk, while the white-noise dataset nullspace has no more overlap than a random subspace. This provides further confirmation that the bulk has a real effect on network behavior, but primarily on inputs that are far out of distribution.

### 3.5 The Bulk is Roughly Independent of Random Seed and Labels

The results of Sec. 3.2 show that along the bulk directions, training behaves linearly across several orders of magnitude. This suggests that the bulk subspaces for independently trained networks may be "similar." To measure the similarity of subspaces, we use the method of principal angles. Specifically, for two subspaces $U \subseteq \mathbb{R}^N$ and $V \subseteq \mathbb{R}^N$ of the same dimensionality, we record the mean of the cosines of the principal angles, yielding an interpretable score in the range $[0, 1]$.[3]

Since there is no hard boundary between the bulk and other subspaces, we want to ensure that our results are not sensitive to any particular threshold. To achieve this, we sort the singular vectors in ascending order by $|\sigma_i - 1|$, and consider the "bulk at $k$" to be the span of the first $k$ vectors in this ordering. We then sweep over a grid of values for $k$, reporting results for each one of them on the x-axis of Figure 5a. As a baseline, we also record the similarity of the bulk at $k$ to a *random* subspace of the same dimensionality.

---

[3]We explored other measures of subspace similarity, such as the geodesic distance on the Grassman manifold, and these did not substantially change the results.

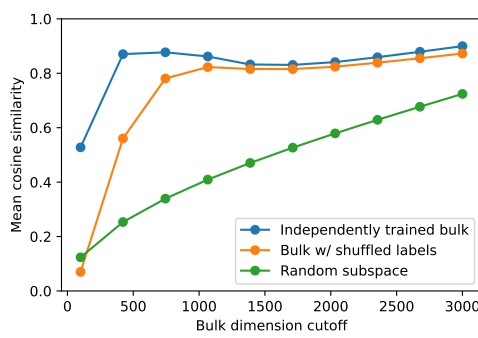 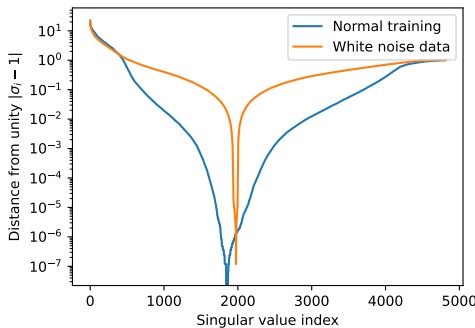

(a) The bulk subspaces for training trajectories starting at two different random initializations are much closer to each other than they are to a randomly sampled subspace of the same dimension.

(b) Replacing the training data with white noise yields a training Jacobian with very few singular values close to one, as compared to training on real data.

Figure 5: The bulk subspaces for training trajectories starting at two different random initializations are much closer to each other than they are to a randomly sampled subspace of the same dimension. The same is true for two trajectories with the same initialization, but where one sees randomly shuffled training labels. By contrast, a trajectory which sees white noise images does not even yield a significant bulk at all.

We find that training trajectories with independent random initializations do have similar bulks, much more similar than one would expect by chance (blue line). We also find that the bulk is not strongly dependent on the training set labels, either (green line).

### 3.6 SGD in a Small Subspace

Following Li et al. (2018) and Song et al. (2024), we examine how the training dynamics of our network change when training is constrained to each of the subspaces we identified above. Specifically, we re-parameterize the network as follows:

$$\theta_t^{(N)} = \theta_0^{(N)} + \mathbf{P}\theta_t^{(n)} \tag{3}$$

where $\mathbf{P}$ is an $N \times n$ projection matrix onto the subspace in question, $\theta_0^{(N)} \in \mathbb{R}^N$ is a fixed random initialization, and $\theta_t^{(n)}$ is vector of learned parameters, initialized with zeros. For ease of comparison, we use identical hyperparameters to those used in Section 1, even though additional epochs may allow training in a restricted subspace to converge further.

Results are plotted in Figure 6. As before, we choose the span of the first $k$ vectors associated with each region and sweep $k$. We choose the $k$ smallest for the stable region, the $k$ largest for the chaotic region and the $k$ closest to 1 for the bulk. We find that when neural network training is constrained to the stable subspace, training is effective for relatively small values of $k$ (around 300), and training on the chaotic subspace is effective for somewhat larger values of $k$ (around 1000). Constraining training to the bulk prevents training from working unless $k$ is very large (around 4000).

## 4 Conclusion

In this work, we introduced a novel lens for understanding neural network training dynamics based on the training Jacobian. We identified a high-dimensional subspace of parameter space, called the bulk, along which the initial parameters are left virtually unaltered by training. The bulk appears to arise from the structure of the input data, largely independent of the labels, and all but disappears when the data lacks structure (e.g. white noise images).

We hypothesize that the structure of the training Jacobian, and especially its bulk subspace, could help explain the inductive biases of neural networks. Our results suggest that the network architecture and input data jointly specify a relatively low-dimensional subspace of parameters which

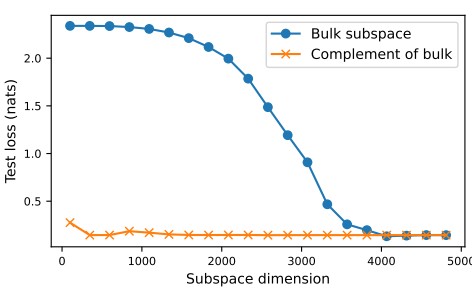 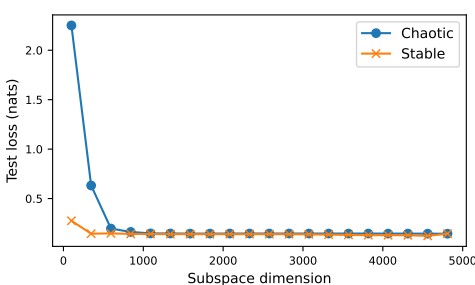

(a) Restricting training to the bulk makes it impossible for SGD to make progress. By contrast, training restricted to the complement of the bulk performs essentially just as well as unconstrained training.

(b) Restricting training to the chaotic subspace (dim $\approx 750$) or to the stable subspace (dim $\approx 1000$) does not significantly harm final test loss.

Figure 6: Restricting training to the bulk subspace makes it impossible for SGD to make progress. By contrast, training restricted to the complement of the bulk performs essentially just as well as unconstrained training.

are "active" during training, leaving the bulk largely untouched. The active subspace seems to be selected in part based on how strongly each direction in parameter space affects in-distribution behavior at initialization (Section 3.4).

Our work is limited insofar as we focus on small-scale models and datasets (see Appendix A.1). Future work should explore ways of analyzing the training Jacobian at a larger scale, perhaps using randomized linear algebra techniques (Yao et al., 2020).

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

# A  APPENDIX

## A.1  LENET-5 RESULTS

In order to validate our results at a larger scale, we computed the full training Jacobian for a LeNet-5 network, consisting of two convolutional layers and three fully connected layers, on the MNIST dataset (LeCun et al., 1998). We chose to slightly modernize the architecture by replacing sigmoid with the ReLU activation function. We used the Adam optimizer with a base learning rate of $0.01$ and a cosine LR decay schedule over three epochs. We did not use any data augmentation or regularization.

The network has a total of $61,706$ trainable parameters, over ten times more than the MLPs we used in our main experiments. The training Jacobian therefore has over 3.8 billion elements, which is sufficient to cause SVD routines in standard machine learning frameworks to crash. To circumvent this issue, we used NumPy's CPU-based SVD implementation, which uses 64-bit indexing internally. This took hundreds of CPU-days just to compute the singular values with `np.linalg.svdvals`, and we were unable to compute the left and right singular vectors in time for the first draft of this paper. We plan to have the full SVD available for the LeNet Jacobian, along with the experiments that it will enable, in time for the next draft.

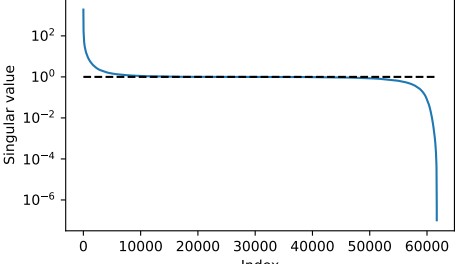
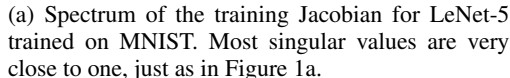
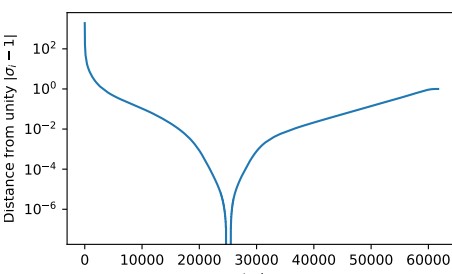

(a) Spectrum of the training Jacobian for LeNet-5 trained on MNIST. Most singular values are very close to one, just as in Figure 1a.

(b) Closer look at how far each singular value is from unity, on a log scale.

Figure 7: LeNet-5 results.