# OpenReview forum: "Understanding Gradient Descent through the Training Jacobian"
_ICLR.cc/2025/Conference — ICLR 2025 Conference Withdrawn Submission_

### Official Review · Reviewer_pXxS · 2024-11-01

**Soundness:** 1
**Presentation:** 4
**Contribution:** 1
**Rating:** 3
**Confidence:** 4

**Summary:**

The authors present some experimental results on a shallow neural network to investigate the behaviour of parameters through training. They show that there seem to exist three (local) subspaces of parameters which behave in distinct ways: those which vary strongly during training, those which remain largely invariant, and those which appear to be damped away by the training algorithm.

**Strengths:**

- The question is undoubtedly an interesting one; understanding the behaviour of parameters during optimisation is important for designing better methods.
- The authors present their findings in a very clear and engaging way.

**Weaknesses:**

I am unfortunately not convinced of the strength of the findings. My major concerns are as follows:
- There appears to be quite a simple explanation for the main findings. The authors themselves show that the 'parameter function Jacobian' has a very similar structure in its singular values, and it seems entirely to be expected that the Jacobian structure will manifest itself during gradient descent. If the networks predictions are indifferent to parameter changes in a given direction (as measured through the Jacobian), then gradient descent will naturally not pick that as a direction in which to optimise.
- The evidence is apparently limited to a single one-hidden-layer perceptron, trained to solve a single relatively simple task (the simple model achieved 'near-zero training loss' in 25 epochs). It is difficult to deduce the existence of some general low dimensional behaviour from this experiment alone.

**Questions:**

- Is there some novel deeper theoretical justification for the authors claims of new low dimensional structure during training?

---

### Official Review · Reviewer_iEL2 · 2024-11-03

**Soundness:** 1
**Presentation:** 3
**Contribution:** 1
**Rating:** 3
**Confidence:** 4

**Summary:**

This papers studies the Jacobian throughout the training trajectory, through the linearization of training, on a small model and dataset. It identifies the bulk of the Jacobian, a subspace of eigenvalues with interesting properties. Gradients in this subspace, however, are shown to be largely unproductive toward improving model performance, which suggests the focus of study toward other 'active' subspaces, and a relationship to the inductive bias of networks with this 'bulk' subspace.

**Strengths:**

The study of training through the 'training Jacobian' appears novel, with many creative experimental methodologies to extract novel relationships between this Jacobian with 'importance' of eigenvectors and with the predictions.
Good presentation with interesting experiments and clear commentary.
The insights drawn are computed largely from properties of the network at initialization, which can be powerful to reveal insights of model performance without costly training.

**Weaknesses:**

The study is limited to a very small set of model/architecture/task combinations.
While the demarcated subspaces of bulk, chaotic, and stable are loose, the additional phase changes observed in the experiments (e.g. Fig 3, Fig 5, Fig 6 are less clear. The empirical evidence suggests the existence of further structure, where some 'thresholds' appear to be taken to match the evidence with the 3-region characterization provided.
While the insights are novel and interesting, the relevance to practical improvements to deep learning, e.g. through generalization, feature learning, and/or inductive biases, are unclear.

**Questions:**

1. Do the small singular values in Fig 4. roughly match with the small singular values in the training Jacobian? It is very surprising and novel that using SGD on the 'stable' eigenvectors can lead to good performance with ~300 eigenvectors, while 'chaotic' eigenvectors form a subspace that is harder to optimize.
2. What do you mean when it says that the structure of the bulk subspace of the training Jacobian could help explain the inductive bias of neural networks?

**Details Of Ethics Concerns:**

No ethical concerns.

---

### Official Review · Reviewer_co1v · 2024-11-04

**Soundness:** 1
**Presentation:** 1
**Contribution:** 1
**Rating:** 3
**Confidence:** 4

**Summary:**

The paper presents an analysis of the geometry of neural network training by examining the training dimensionality through the lens of linearizing training parameters around an initialization point. While the authors provide an interesting perspective by differentiating through the training process and analyzing the Jacobian matrix of final parameters relative to initial ones, several critical issues warrant rejection.
The study's primary focus on the spectral analysis of the Jacobian matrix and its relationship to singular values appears to reiterate concepts already well-established in existing literature. For instance, previous works have extensively discussed the implications of singular value decomposition (SVD) in neural networks and their role in understanding gradient behavior and stability during training. The claim that singular values can be categorized into distinct regions (bulk, stable, chaotic) lacks sufficient novelty, as similar categorizations have been previously documented without substantial new insights or methodologies.
Although the authors assert that their empirical findings support their claims about the stability of the bulk region during training, they fail to provide robust experimental evidence or a comprehensive evaluation across diverse network architectures and datasets. The reliance on a narrow set of experiments raises concerns about the generalizability of their conclusions. A more thorough exploration that includes various neural network configurations and training scenarios would strengthen their argument.
The hypothesis that the unchanged bulk region may elucidate the inductive bias concept in neural networks is intriguing but remains inadequately supported. The authors do not sufficiently connect their empirical observations to established theories of inductive bias, leaving a gap in theoretical justification. A more rigorous theoretical framework is necessary to substantiate such claims.
In conclusion, while the paper addresses an interesting aspect of neural network training dynamics, its lack of novelty, insufficient empirical validation, and weak theoretical grounding render it unsuitable for publication in its current form. Further development addressing these concerns would be required for reconsideration.

**Strengths:**

1)	The paper demonstrates empirically three different regions of the singular value spectrum of the training Jacobian around initialization. This it is heavily explored research direction and this paper provides further empirical evidence on the training characterization of shallow NNs, including random feature models and two-layer NNs with lazy training regime.
2)	They further report that the structure of the input data plays a key role (largely independent of labels) aligning with prior results on isotropic vs. anisotropic data models.

**Weaknesses:**

1)	The current work addresses a specific direction of exploring the neural network training via the spectral analysis of the Jacobian of the trained parameters with respect to initialization. This regime has been already shown to perform equivalent to linear models, and the literature already focus by far on alternative methods that enable to overperform linear models. Hence, the motivation of the current work and how it interplays with rest of the literature lacks fundamental  details.
2)	The structure of the input data (possibly low-dimensional) and different learning rates that scale with the dimensionality are known to be some alternative ways to achieve better than the linear regime in the literature. In the current paper, authors claim that network architecture and the input data structure jointly specify a low-dimensional subspace which partly stays active or left untouched during training. The justification of this point with further evidence either theoretically or empirically would be necessary.

**Questions:**

1)	The experimental details lacks information about the architecture of the MLP, the choice of the dimension of the hidden layer, the nonlinearity etc.
2)	The current way to compute Jacobian and perform SVD on it limits its applicability on small dimensions. The authors reported that it took hundreds of CPU days for the LeNET-5 results in appendix. Hence, the motivation behind this approach needs further explanation.

---

### Official Review · Reviewer_dx4x · 2024-11-04

**Soundness:** 3
**Presentation:** 2
**Contribution:** 2
**Rating:** 5
**Confidence:** 3

**Summary:**

This paper studies the properties of the Jacobian of trained parameters relative to their initial values to better understand an intrinsic low-dimensional structure in the training dynamics. In particular, the authors reveal three different regions by analyzing the spectral properties of the Jacobian and investigate the characteristics of the 'bulk' region (region associated to the singular values close to one).

**Strengths:**

The paper is well written.

The subject is interesting and this paper sheds light on insightful phenomena.

**Weaknesses:**

- the code is not available

- The bibliography part seems quite light (10 cited papers)

- The authors need to formalize some concepts rigorously. For example, the 'dimensionality of training' is not defined, though the notion seems clear for the authors as they claim "Clearly (...) the dimensionality of training is equal to (...)" ll. 089, 090.

- I did not understand the part l. 092-094, so I would like to have more precise explanations.

- typos: "to the the Jacobian of" l. 240

**Questions:**

- By assuming equation (1), is it not equivalent to assuming that we are in the gradient flow limit?

- what happens during the training for the spectral analysis of the Jacobian, do the authors observe the same phenomenon as in Figure 1, or is it only at the end of the training? And if there is the same phenomenon during training, do the indexes associated with the singular values close to one remain the same (I mean without previously sorting them)?

- is Figure 1 consistent for different initializations? Does Part 3.2 suggest that the indexes associated to the "bulk" are the same?

- what happens for all experiments if you do not use a momentum term? As there are conserved quantities (independent of the data set) without momentum and none with momentum and as conserved quantities force the dynamic to stay in a lower dimensional manifold, the comparison between the two optimizations should show some interesting phenomena.

---

### Official Review · Reviewer_6eiD · 2024-11-05

**Soundness:** 2
**Presentation:** 3
**Contribution:** 1
**Rating:** 3
**Confidence:** 3

**Summary:**

This paper studies the training dynamics of neural networks through the Jacobian of the function at initialization. The paper shows that the singular value spectrum of the Jacobian consists of three distinct regions. The "chaotic", "bulk" and "stable" regions. The empirical results in the paper suggest that there is a low-dimensional structure in the training process of deep neural networks which depend on the input data but are largely independent of the labels.

**Strengths:**

- Understanding the training dynamics of deep neural networks is interesting and can have lots of potential in accelerating both training and inference of these large models.
- The paper is well written and most of the figures are clear.

**Weaknesses:**

- The biggest weakness is that the experiments are carried out on extremely simple datasets and models. Therefore it is unclear whether the conclusions being made can apply to modern deep learning architectures and datasets. I would suggest the authors either provide some theoretical guarantees or conduct experiments on more realistic models/datasets.

- The other major weakness is that it is hard to understand how this work fits into the current body of literature and what new insights it brings. The authors only mention 3 works related to the idea being studied in this paper. My understanding is that there is a large body of literature attempting to understand the training dynamics of neural networks through some "low dimensional subspace". For example [1,2] and references therein. There also seems to be an implicit assumption be made that the parameters do not move far from initialization so I am also curious how this relates to the NTK/lazy training regime [3,4].

Overall, the paper feels very incomplete and not ready for publication. I would encourage the authors to conduct larger experiments and provide a more detailed explanation for how their results fit into our current understanding of neural network training.

**Questions:**

- Where is the dashed line in Figure 2a? It's not clear.

[1]Kwon, Soo Min, et al. "Efficient low-dimensional compression of overparameterized models." International Conference on Artificial Intelligence and Statistics. PMLR, 2024.

[2] Yaras, Can, et al. "Compressible Dynamics in Deep Overparameterized Low-Rank Learning & Adaptation." arXiv preprint arXiv:2406.04112 (2024).

[3] Jacot, Arthur, Franck Gabriel, and Clément Hongler. "Neural tangent kernel: Convergence and generalization in neural networks." Advances in neural information processing systems 31 (2018).

[4] Chizat, Lenaic, Edouard Oyallon, and Francis Bach. "On lazy training in differentiable programming." Advances in neural information processing systems 32 (2019).

---

### Note · Authors · 2024-11-21

**Comment:**

We thank the reviewers for their helpful feedback. We will work to improve the paper before submitting to a future conference.

**Withdrawal Confirmation:**

I have read and agree with the venue's withdrawal policy on behalf of myself and my co-authors.